# Deficiency in MT5-MMP Supports Branching of Human iPSCs-Derived Neurons and Reduces Expression of GLAST/S100 in iPSCs-Derived Astrocytes

**DOI:** 10.3390/cells10071705

**Published:** 2021-07-06

**Authors:** Nikita Arnst, Pedro Belio-Mairal, Laura García-González, Laurie Arnaud, Louise Greetham, Emmanuel Nivet, Santiago Rivera, Alexander Dityatev

**Affiliations:** 1Molecular Neuroplasticity, German Center for Neurodegenerative Diseases (DZNE), 39120 Magdeburg, Germany; arnst.nikita@yandex.ru (N.A.); pedro.BELIO-MAIRAL@univ-amu.fr (P.B.-M.); 2Inst Neurophysiopathol, CNRS, INP, Aix Marseille Université, 13385 Marseille, France; laura.garcia-gonzalez@outlook.es (L.G.-G.); laurie.ARNAUD@univ-amu.fr (L.A.); louise.GREETHAM@univ-amu.fr (L.G.); Emmanuel.NIVET@univ-amu.fr (E.N.); santiago.rivera@univ-amu.fr (S.R.); 3Center for Behavioral Brain Sciences (CBBS), 39106 Magdeburg, Germany; 4Medical Faculty, Otto-von-Guericke University, 39120 Magdeburg, Germany

**Keywords:** Alzheimer’s disease, human-induced pluripotent stem cells, disease modeling, neuronal differentiation, hiPSC-derived astrocytes, metalloproteinase, morphometry, whole-cell patch-clamp

## Abstract

For some time, it has been accepted that the β-site APP cleaving enzyme 1 (BACE1) and the γ-secretase are two main players in the amyloidogenic processing of the β-amyloid precursor protein (APP). Recently, the membrane-type 5 matrix metalloproteinase (MT5-MMP/MMP-24), mainly expressed in the nervous system, has been highlighted as a new key player in APP-processing, able to stimulate amyloidogenesis and also to generate a neurotoxic APP derivative. In addition, the loss of MT5-MMP has been demonstrated to abrogate pathological hallmarks in a mouse model of Alzheimer’s disease (AD), thus shedding light on MT5-MMP as an attractive new therapeutic target. However, a more comprehensive analysis of the role of MT5-MMP is necessary to evaluate how its targeting affects neurons and glia in pathological and physiological situations. In this study, leveraging on CRISPR-Cas9 genome editing strategy, we established cultures of human-induced pluripotent stem cells (hiPSC)-derived neurons and astrocytes to investigate the impact of MT5-MMP deficiency on their phenotypes. We found that MT5-MMP-deficient neurons exhibited an increased number of primary and secondary neurites, as compared to isogenic hiPSC-derived neurons. Moreover, MT5-MMP-deficient astrocytes displayed higher surface area and volume compared to control astrocytes. The MT5-MMP-deficient astrocytes also exhibited decreased GLAST and S100β expression. These findings provide novel insights into the physiological role of MT5-MMP in human neurons and astrocytes, suggesting that therapeutic strategies targeting MT5-MMP should be controlled for potential side effects on astrocytic physiology and neuronal morphology.

## 1. Introduction

Alzheimer’s disease (AD) is a neurodegenerative brain disorder with no efficient cure to date, which is typically characterized by a progressive cognitive decline [1,2]. Many *in vitro* and *in vivo* studies have highlighted the BACE1-dependent abnormal formation and intracellular accumulation of amyloid β (Aβ) peptides as the initial cause of AD [3,4]. Therefore, BACE1 has been one of the most attractive targets for the treatment of AD for many years. Accordingly, multiple BACE1 inhibitors (MK8931, AZD-3293, JNJ-54861911, E2609, and CNP520) have been tested as potential therapeutic drugs to treat neurological diseases [5,6,7]. However, the high failure rate of clinic-step BACE1 inhibitor candidates and their interference with the physiological processing of other substrates has stimulated searching for novel therapeutic targets in AD [8,9,10].

Matrix metalloproteinases (MMPs), enzymes controlling intracellular and extracellular amyloidogenic protein turnover, are associated with neurodegenerative diseases [11,12]. Recently, we shed light on MT5-MMP as just such a novel therapeutic target [12,13,14]. In good health, MT5-MMP is a multifunctional MMP with N-cadherin substrate specificity that is mainly expressed in developmental stages and remains high in adulthood in the hippocampus and cerebellum [15,16,17,18]. It may play a functional role in neural plasticity via interactions with the AMPA binding protein (ABP) and the glutamate receptor-interacting protein (GRIP) [19,20]. Besides, the localization of the metalloproteinase in filopodia at the tips of growth cones and in synaptosomes also supports the idea that MT5-MMP may regulate axon pathfinding and synapse remodeling [21,22]. MT5-MMP also regulates the activation of adult neural stem cells [23]. However, MT5-MMP is also relevant to disease, since it promotes interactions between neurons and mast cells through the cleavage of N-cadherin, required for the inflammatory response in a thermal pain model [24]. In addition, MT5-MMP was found around senile plaques in the post-mortem brains of AD patients [15]. Furthermore, the physical interaction of MT5-MMP with APP and increased production of Aβ/C99 also corroborate the idea that the proteinase is a new, full-fledged player in APP metabolism, and hence a potential new target molecule in AD treatment [25,26]. Finally, *in vitro* and *in vivo* studies revealed a novel physiological APP processing pathway, the product of which is a proteolytic fragment Aη-α capable of reducing long-term potentiation in hippocampal neurons in the CA1 pyramidal cell layer [27]. 

Analysis of bigenic mice, resulting from crossing the 5xFAD mouse model of AD and MT5-MMP knockout mice, revealed several positive effects of MT5-MMP deficiency at early stages of the pathology, compared to age-matched 5xFAD. These include the prevention of deficits in long-term potentiation and learning and memory, decreased levels of Aβ/C99, interleukin-1β (IL-1β) and tumor necrosis factor-α (TNF-α), as well as decreased glial reactivity [25,26]. Interestingly, MT5-MMP knockout mice showed no apparent phenotypes until the organism was subjected to stressful conditions [3,23]. In this vein, MT5-MMP knockout mice do not develop neuropathic pain, together with the absence of Aβ-fiber sprouting after sciatic nerve injury [21], and they do not develop thermal hyperalgesia during inflammation [24]. Besides, confocal imaging and qRT-PCR analyses of samples after traumatic brain injury (TBI) and bilateral entorhinal cortical lesion revealed that MT5-MMP inhibition attenuated ADAM-10 and N-cadherin [19].

Overall, MT5-MMP appears to be a pivotal enzyme in multiple physiological and pathological pathways that affect both neurons and glial cells. Accordingly, we investigated the influence of the MT5-MMP knockout state on the phenotype of human neurons and astrocytes and their possible functional crosstalk in a non-pathological situation. Using differentiated human neural derivatives from hiPSCs and CRISPR/Cas9 approaches, we showed that MT5-MMP deficiency led to changes in neuron and astrocyte morphology, characterized by the formation of more branched neurons and round-shaped astrocytes with decreased expression of S100b and GLAST. These data corroborate the multifaceted spectrum of MT5-MMP actions in the nervous system.

## 2. Materials and Methods

### 2.1. Cell Line

A commercially available human cell line was purchased from the Coriell biorepository, and originate from the CIRM iPSC Repository, under the reference CW70019. The CW70019 hiPSC line was obtained from a donor being non-demented at the time of biopsy.

### 2.2. hiPSC Culture

hiPSCs were cultured and maintained undifferentiated in a chemically defined growth medium (StemMACS™ iPS-Brew XF media; Miltenyi Biotec, Paris, France) onto growth-factor-reduced Matrigel (BD Biosciences, Franklin Lakes, NJ, US)-coated plates (8.6 μg/cm^2^). In short, when reaching 70–80% confluency, hiPSCs were treated with an enzyme-free solution (hereafter referred to as gentle dissociation solution) containing 0.5 mM EDTA (Lonza, Basel, Switzerland), D-PBS (Gibco Life Technologies, Scotland, UK) and 1.8 mg/mL NaCl (Sigma-Aldrich, st Louis, MO, USA). hiPSCs were incubated for 2 min in gentle dissociation solution at 37 °C, and the colonies were dispersed to small clusters and lifted carefully using a 5 mL glass pipette at a ratio of 1:3 or 1:4 for further amplification. When necessary, differentiated areas were removed from hiPSC cultures prior to passaging, in order to maintain the cultures as undifferentiated before proceeding to their differentiation. hiPSC lines were maintained in an incubator (37 °C, 5% CO_2_) with daily medium changes.

### 2.3. CRISPR-Cas9 Plasmid Production

For the generation of CRISPR-Cas9 plasmids, we first modified the pSpCas9(BB)-2A-Puro (PX459) V2.0 (Addgene plasmid # 62988, Addgene, Teddington, UK) [28] to replace the CBh promoter with the EF1 alpha promoter by a subcloning strategy. The modified plasmid was named cAB03 (i.e., pX459-pEF1 alpha), and further used to produce two distinct plasmids, each containing a single guide RNA specific to the *MMP24* sequence (i.e., sgRNA containing crRNA + TracrRNA), aiming at generating insertions and deletions within the exon 2. To that end, CRISPR sgRNAs were designed using the Zhang lab design tool (http://CRISPR.mit.edu, accessed on 28 June 2021) and cloned into cAB03 as previously described [28]. For the sgRNA, a 20 nt *MMP24* specific sequence was selected, sgRNA1: GATCCCGGTCACCGGTGTGT. Briefly, the selected *MMP24* specific sequence was cloned by BbsI-mediated digestion of cAB03, followed by ligation using cohesive end cloning. Plasmids were transformed into DH5α chemically competent cells (Thermo Fisher Scientific, Waltham, MA, USA) and individual clones were first screened by PCR to verify the insertion, then were ultimately validated by sequencing analysis.

### 2.4. CRISPR-Cas9-Mediated Edition for the Generation of MMP24 Knockout hiPSCs

To generate *MMP24* knockout hiPSCs from the CW70019 hiPSC line, cells were prepared and electroporated as previously described [29]. After clonal isolation, for each clone DNA, extraction was performed using a thermocycler with the following settings: 3 h at 55 °C and 30 min at 95 °C. Then, a 10 µL PCR reaction was performed using 1 µL of the extracted genomic DNA and 9 µL of a PCR mix containing 1× of the 5× PrimeSTAR GXL buffer (Takara Bio, Shiga, Japan), PrimeSTAR GXL DNA polymerase (0.25 U/10 µL), dNTPs (200 µM each) and the following primers: *MMP24*For 5′-ACGGGCATCTGCGCTGCACTC-3′ and *MMP24*Rev 5′-GCTGGGTGCCTGCATGTGCCTG-3′ (0.3 µM each). PCR reaction was performed using the following conditions: 1 min at 94 °C for one cycle, then 10 sec at 98 °C and 1 min at 55 °C for 35 cycles, and to finish, 5 min at 68 °C for one cycle. Upon completion, the PCR product was digested using the restriction enzyme AleI (New England Biolabs, Ipswich, MA, US), and run on a 2% agarose gel electrophoresis to analyze the size of the amplicon for each individual clone (expected profiles: two bands for the wild-type clone at 466 bp and 209 bp; one band for the homozygous clone at 675 bp; three bands at 675 bp, 466 bp and 209 bp for the heterozygous clone). Only knockout clones for *MMP24* were kept and further confirmed by Sanger sequencing prior to amplification for further experiments. Three knockout clones were produced for the experiment.

### 2.5. Derivation of NPCs from hiPSCs

hiPSCs were differentiated into neural progenitor cells (NPCs) following a monolayer culture method with a commercial dual SMAD inhibition-mediated neural induction medium (STEMdiff™ SMADi Neural Induction Kit, Stem Cell Technologies, Vancouver, Canada), based on a previously published protocol [30]. NPCs were obtained following the manufacturer’s instructions, with slight modifications. Undifferentiated cultures of hiPSCs were treated with gentle dissociation solution for 4 min, then the solution was removed and cells incubated in Accumax for 4 min. hiPSCs were then dislodged as single cells and transferred into Dulbecco’s modified Eagle’s medium (DMEM)-F12 + 20% knockout serum replacement. Then, cells were centrifuged at 200× *g* for 5 min, resuspended in D-PBS, counted and centrifuged once more at 200× *g* for 5 min. Lastly, cells were resuspended in STEMdiff™ SMADi neural induction kit + 10 µM Y-27632 (Tocris Bioscience, Bristol, UK), a Rho-associated protein kinase inhibitor, plated onto growth-factor-reduced Matrigel™ (BD Biosciences, Franklin Lakes, NJ, USA)-coated plates (8.6 μg/cm^2^) at 320,000 cells/cm^2^, and maintained in an incubator (37 °C, 5% CO_2_) with medium changes every day. Between 6 and 9 days after induction, cells were harvested as single cells using Accumax, transferred into DMEM-F12 medium, centrifuged at 200× *g* for 5 min, resuspended in STEMdiff^TM^ SMADi medium + 10 μM Y-27632, plated onto growth-factor-reduced Matrigel-coated plates (8.6 μg/cm^2^) at 270,000 cells/cm^2^ and maintained in an incubator (37 °C, 5% CO_2_) with medium changes every day for 5 additional days. Then, cells were passaged one more time following the same procedure. Five days after the last passage (i.e., between 16 to 19 days post-induction), differentiated cells were harvested as single cells using Accumax, centrifuged at 200× *g* for 5 min, before being resuspended in STEMdiff^TM^ neural progenitor medium (Stem Cell Technologies, Vancouver, Canada; hereafter referred as NPC media) and seeded onto growth-factor-reduced Matrigel-coated plates (8.6 μg/cm^2^) at 125,000 cells/cm^2^. hiPSC-NPCs were maintained in an incubator (37 °C, 5% CO_2_) with medium changes every day, and passaged with Accumax when reaching 80–90% confluency.

### 2.6. Neuronal Differentiation

hiPSC-derived NPCs were differentiated in accordance with a previously published protocol [31]. In brief, hiPSC-NPCs were detached using Accutase, collected, centrifuged, and gently re-suspended in NPC medium and plated onto poly-L-ornithine (PLO) (Sigma-Aldrich, st Louis, MO, USA; P3655; 15 µg/mL)/Laminin (Life Technologies, Waltham, MA, USA, 23017015; 10 µg/mL)-coated glass coverslips at a density of 50,000 cells/cm^2^ and incubated overnight (37 °C with 5% CO_2_). Thereafter, the NPC medium was removed and replaced by the BrainPhys^TM^ neuronal media kit (Stem Cell Technologies, Vancouver, Canada) composed of the BrainPhys neuronal maturation medium (StemCell Technologies, Vancouver, Canada; #05793) supplemented with NeuroCult SM1 neuronal supplement (StemCell Technologies, #05711) + N2 Supplement-A (StemCell Technologies, #07152) + 20 ng/mL BDNF (Peprotech, Rocky Hill, NJ, USA; #450-02) + 20 ng/mL GDNF (Peprotech; #450-10) + 1 mM dibutyryl-cAMP (Sigma-Aldrich; D0627) + 200 nM ascorbic acid (Sigma-Aldrich; A5960). Half of the medium was gently replaced two to three times a week for 8 weeks. The plates were kept in a humified incubator at 37 °C with 5% CO_2_. In all experiments, the passage number (P7–P9) and the time in culture (8 weeks) were identical.

### 2.7. Astrocyte Differentiation

Differentiation of hiPSC-derived NPCs into astrocyte-like cells was performed using a previously published protocol with slight modifications [32]. In short, hiPSC-NPCs were dissociated with Accutase™ and seeded at 30,000 cells/cm^2^ on growth-factor-reduced Matrigel™-coated 6-well plates (8.6 μg/cm^2^) in astrocyte medium (ScienCell, Carlsbad, CA, USA, 1801-b), containing 2% fetal bovine serum (0010), astrocyte growth supplement (1852) and 10 U/mL penicillin/streptomycin solution (0503). Full medium changes were performed every 72 h over a 30-day differentiation period. When cells reached 90–95% confluence (approximately 5–7 days after initial seeding), they were passaged as single cells into astrocyte medium and cultured onto growth-factor-reduced Matrigel™-coated plates (8.6 μg/cm^2^) at 30,000 cells/cm^2^. The cells were plated onto a non-coated plastic surface after 21 days (around 3–4 passages) to remove immature astroglial cells. The cells were used for further experiments after 45 days of incubation in astrocyte medium. The maturity of astrocytes was evaluated with immunocytochemistry (ICC) for ALDH1L1, S100β, GALST, and GFAP. In all experiments, the passage number (P7–P9) and the time in culture (45 DIV) were identical.

### 2.8. Immunocytochemistry

The cells were fixed with a 4% paraformaldehyde solution (PFA) (Sigma-Aldrich; 158127) in PBS (Gibco, Scotland, UK) pH 7.4, for 10 min on a bed of ice. Thereafter, the cells were incubated in a quenching solution (0.1 M aqueous glycine solution) for 5 min and washed with PBS, pH 7.4, for 5 min. The fixed cells were then incubated in a blocking solution (5% normal goat serum (Jackson ImmunoResearch, West Grove, PA, USA), 0.1% Triton X-100 (Sigma-Aldrich; 2315025) diluted in PBS, pH 7.4 for 1 h at RT. The cells were then incubated with the following primary antibodies diluted in blocking solution overnight with gentle shaking at 4 °C: SOX2 (1:1000, Abcam, Cambridge, UK; ab97959), PAX6 (1:1000, BioLegend, San Diego, CA, USA; 901302), Nestin (1:500, Merck Millipore, Burlington, MA, USA; MAB5326), NeuN (1:500, Merck Millipore; MAB377), S100β (1:100, Abcam; ab212816), vGluT1 (1:1000, Merck Millipore; AB5905), PSD-95 (1:500, Abcam; ab12093), MAP2B (1:500, Abcam; ab5392), GLAST (1:120, Thermo Fisher Scientific; PA5-34198), GFAP (1:100, Merck Millipore; AB5541), and ALDH1L1 (1:100, Abcam; ab190298). The next day, the cells were thoroughly washed three times with 0.1% PBS-Triton X-100 and incubated with the following species-specific fluorochrome-conjugated secondary antibodies diluted in PBS (1:1000) for one hour at RT: Alexa Fluor 405 anti-rabbit IgG (Abcam; ab175651), Alexa Fluor 405 anti-mouse IgG (Life Technologies; A-31553), Alexa Fluor 405 anti-chicken IgG (Life Technologies; A-11040), Alexa Fluor 488 anti-mouse IgG (Life Technologies; A-11029), Alexa Fluor 546 anti-mouse IgG (Life Technologies, A-11030), Alexa Fluor 546 anti-rabbit IgG (Life Technologies; A-11035), Alexa Fluor 568 anti-rabbit IgG (Life Technologies; A-11036), and Alexa Fluor 633 anti-guinea pig IgG (Life Technologies; A-21105). After rinsing three times with 0.1% PBS-Triton X-100, pH 7.4, the samples were counterstained with the nuclear marker DAPI. Finally, coverslips were rinsed three times with PBS, one more time with phosphate buffer (pH 7.4) and mounted with a mounting medium (Sigma-Aldrich; M1289). Microscope slides with mounted coverslips were stored at 4 °C until the next morning, when the mounting medium had dried. Then, the edges of coverslips were treated with nail polish to prevent the samples from drying out. The rest of the time, all samples were stored at 4 °C (no more than 4 days) until the imaging of samples was completed.

### 2.9. Confocal Microscopy and Data Collection

Serial optical z-sections were carefully collected at 0.28 μm intervals using a Zeiss Axio Imager 2 (Carl Zeiss GmbH, Jena, Germany), in combination with a Carl Zeiss LSM 700 system equipped with an EC “Plan-Neofluar” 40×/0.75 M27 and EC “Plan-Neofluar” 63×/1.40 Oil DIC M27 objectives, and 405 nm, 488 nm, 555 nm and 633 nm laser lines. Throughout all experiments, the pinhole was set to 0.5 Airy unit. The images were collected using a bidirectional scan and two frames’ averages settings. The z-sections were deconvolved using a fast iterative algorithm (five iterations) integrated into Zeiss software. All images in this work were maximum intensity projections, which were analyzed using Fiji (ImageJ, NIH, USA, 1.53c) or/and Imaris software (Bitplane AG, Zurich, Switzerland, v.7.7.2). 

### 2.10. Image Analysis

Fiji Sholl analysis (v.3.6.12) was utilized to describe neuronal arbors. For analysis, the scale-corrected maximum intensity z-stack projection images of a single neuron were obtained with EC “Plan-Neofluar” 63×/1.40 Oil DIC M27 objectives. Thereafter, the images were converted into an 8-bit grayscale format. The thresholding option was utilized to obtain a mask of the neuron. The Otsu thresholding algorithm (v.1.17.2) was consistently used throughout the Sholl image analysis. The neurites of neighboring neurons were removed with the eraser tool. Using a “Line segment tool”, a single line from the center of the soma of a neuron to its longest neurites was drawn. The concentric shell rings with 5 μm steps were drawn automatically. The first shell was defined 10 μm outside of the cell body to exclude the latter from the analysis. Thereafter, a repeated two-way ANOVA was performed.

Quantification and comparison of the neurite structures were achieved with the NeuronJ plugin (v.1.4.3) for Fiji. For analysis, the maximum intensity projection two-dimensional (2D) images were converted into 8-bit gray-scale type and loaded through the NeuronJ toolbar. Thereafter, the neurite structures were manually traced and binned into two groups, referred to as “primary” and “secondary” neurites, for further automatic analysis. 

To quantify the GFAP cytoskeleton volume and surface area, images were subsequently analyzed with the Imaris software (Bitplane AG, Zurich, Switzerland) using the “Surface tool”, as described elsewhere [33]. After that, the ratio between the surface area and volume was calculated using Excel.

Analysis of fluorescent intensity of astroglia markers was done with ImageJ, as described elsewhere [33]. Shortly, the cells from five random fields of three random wells of three independent differentiations were chosen for analysis. After deconvolution, the average background fluorescence, calculated from three random regions of each field of view, was subtracted. For comparisons, we used the value of mean integrated density per cell, which was measured as raw integrated pixel density divided by the number of immunoreactive cells.

### 2.11. Patch-Clamp Recordings

Whole-cell patch-clamp recordings of neurons were performed after 8 weeks in the absence of any receptor antagonists, as described previously [31]. The coverslips with neurons were transferred into a heated perfusion chamber with a constant flow (1 mL/min) of warm artificial cerebrospinal fluid (ACSF: 121mM NaCl, 4.2 mM KCl, 1.0 mM MgSO_4_, 0.45 mM NaH_2_PO_4_, 0.5 mM Na_2_HPO_4_, 29 mM NaHCO_3_, 1.1 mM CaCl_2_, 20 g glucose, all from Sigma-Aldrich), bubbled with carbon dioxide and maintained at 37 °C. Cells were incubated at 37 °C in a perfusion chamber for 5 min before recordings, to minimize the “transfer” effects. Patch pipettes were pulled from borosilicate glass capillaries and had a resistance of 5–8 MΩ (in the whole-cell configuration). Patch electrodes were filled with internal solution containing 130 mM K-gluconate, 6 mM KCl, 4 mM NaCl, 10 mM Na-HEPES, 0.2 mM K-EGTA, 0.3 mM GTP, 2 mM Mg-ATP, 0.2 mMcAMP, and 10 mM D-glucose (all from Sigma-Aldrich). The pH and osmolality of the internal solution were 7.3 and 290 mOsm, respectively. Currents were recorded with an EPC9 (HEKA Elektronik, Lambrecht, Germany) patch-clamp amplifier. Electrode capacitances were compensated online in the cell-attached mode. We maintained the cells in voltage clamp (VC) at −70 mV. Voltage-dependent Na^+^ and K^+^ channel properties were examined in response to stepwise membrane depolarization, and the IV curves were computed. Spontaneous synaptic events were recorded for at least 3 min at −70 mV (close to the Cl^−^ reversal potential, suggesting that the recorded events were EPSCs). Then we switched to current-clamp mode to evaluate the ability of cells to generate action potentials. Data acquisition and quality were controlled in real-time by PatchMaster software (HEKA Elektronik, Lambrecht, Germany) and saved for further analysis.

### 2.12. Statistical Analysis

All statistical analyses were performed using GraphPad Prism version 9.0 for Windows (GraphPad Software, San Diego, CA, USA). Each experiment was performed at least three times. All the data are presented as mean ± S.E.M. Results were considered significant if *p* < 0.05. The results of Sholl analysis were evaluated with a two-way repeated-measures ANOVA, followed by Bonferroni’s multiple comparison test (genotype/distance from the soma). Neuronal morphology analysis was achieved with a nested *t*-test. Data were analyzed by Shapiro–Wilks’s test to determine normality. Thereafter, the log transformation was used if the data were not distributed normally. Astrocyte morphology data were analyzed with both two-way ANOVA (genotype/morphotype) and nested *t*-test (the same morphotype comparison between WT and MT5-MMP KO). Mean fluorescence intensity per cell data were analyzed with a nested *t*-test (the same morphotype comparison in both WT and MT5-MMP KO). The data collection was not genotype-blinded.

## 3. Results

### 3.1. MT5-MMP Deficient hiPSCs Preserve the Ability to Differentiate into Neuronal Cells

To accurately assess the impact of MT5-MMP deficiency on human neural cells, we first decided to knock out the *MMP24* gene from a hiPSC line that was previously generated from a non-demented donor. Upon CRISPR-Cas9 genome editing, hiPSC clones were evaluated for insertions/deletions in the targeted exon 2 (Figure 1A). PCR-based screening (Figure 1B), further validated by Sanger sequencing, allowed us to select and confirm the generation of several isogenic MT5-MMP knockout clones. Then, the loss of MT5-MMP was further validated by genic analyses in those selected isogenic MT5-MMP KO hiPSC clones. Indeed, MT5-MMP mRNA levels were barely detectable in the validated clones by qPCR, which demonstrated CRISPR-Cas9-induced frameshift mutations, leading to functional loss of MT5-MMP (Figure 1C).

In the present study, both wild-type and isogenic MT5-MMP knockout hiPSCs were successfully converted into neural progenitor cells (NPCs). Both cell lines were characterized by the expression of neuroepithelial markers PAX6 (Figure 2A,A’), SOX2 (Figure 2B,B’) and nestin (Figure 2A,B). Microscopic observations revealed neither specific morphological differences nor expression marker changes between control hiPSC-NPCs and their isogenic MT5-MMP knockout counterparts. The neuronal differentiation capacity of MT5-MMP knockout hiPSC-NPCs was similar to that of wild-type cells. After three days of neuronal differentiation onset, multiple neurite-like protrusions were observed. At that early time of the differentiation stage, MT5-MMP knockout cells showed indistinguishable morphology as compared to wild-type cells (Figure 2C,D). Besides, both wild-type and MT5-MMP knockout neuronal-like cells at DIV60 showed comparable expression of neuronal markers NeuN and MAP2B (Figure 2E–H). These findings indicate that MT5-MMP deficiency does not hamper the overall differentiation potential of hiPSCs toward a neural progenitor-like phenotype and its neuronal derivatives. However, initial visual analysis of wild-type and MT5-MMP knockout neurons after 60 days in maturation conditions revealed that neurons exhibited distinguishable morphological features related to their arborizations (Figure 2G,H).

### 3.2. MT5-MMP Deficient Neurons Exhibited Increased Branching of Primary and Secondary Neurites

To describe morphological changes quantitatively, we conducted a morphometric analysis of hiPSC-derived neurons. Using the Sholl analysis, we revealed that MAP2B-positive MT5-MMP-deficient neuronal cells had a significantly greater number of dendritic intersections in the interval ranging between 25 to 50 μm from neuronal soma, compared to MT5-MMP wild-type neurons (two-way repeated-measures ANOVA with Bonferroni’s post hoc test, *N* = 3 culture preparations, *n* = 82 neurons, *F* (1, 80) = 10.7, *p* = 0.002, Figure 3A). Comparative analysis with a nested *t*-test showed that MT5-MMP deficiency resulted in an increased number of primary (*N* = 3, *n* = 111, *F* = 10.0, *p* = 0.03, Figure 3B) and secondary neurites (*N* = 3, *n* = 111, *F* = 42.8, *p* = 0.003, Figure 3C). Next, we investigated whether these changes in neurite numbers were also accompanied by changes in neurite lengths. The average length of primary (*N* = 3, *n* = 111, *F* = 0.0126, *p* = 0.92, Figure 3D) and secondary (*N* = 3, *n* = 111, *F* = 3.30, *p* = 0.14, Figure 3E) neurites was not different between wild-type and MT5-MMP deficient neurons. Predictably, the total length—i.e., an average sum length of all primary and secondary neurites per neuron—was significantly higher for MT5-MMP deficient cells because of the increased number of neurites (*N* = 3, *n* = 111, *F* = 28.0, ** *p* = 0.006, Figure 3F).

### 3.3. MT5-MMP Deficiency Does Not Alter Neuronal Activity in hiPSC-Derived Neurons

Considering the morphological differences observed in MT5-MMP-deficient neuronal cultures, we next sought whether such changes would also be accompanied by functional alterations. The presence of vGLUT1/PSD95 positive synaptic-like structures at 60DIV (Figure 4A,B) indicated that neurons could be involved in functional interactions. Therefore, whole-cell patch-clamp recordings were conducted. Neurons of both genotypes had relatively small intrinsic Na+ and K+ currents (Figure 4C,D) and generated only single-action potentials in response to the depolarization steps (Figure 4E). Nonetheless, some cells were characterized by the presence of inward spontaneous postsynaptic currents (Figure 4F). We observed that the occurrence of postsynaptic currents was limited in the generated hiPSC-derived neuronal cultures, independently of the genotype (wild-type: *n* = 9 cells out of 21; MT5-MMP knockout: *n* = 7 cells out of 23). Therefore, overall, we did not observe genotype-associated effects on the electrophysiological properties at the studied developmental stage. However, we cannot rule out that more mature neurons would display genotype-specific differences in their functionalities.

### 3.4. MT5-MMP Deficiency Induced Morphological Changes and Reduced the Expression of GLAST and S100β in Astrocyte-Like Cells

We next sought to evaluate MT5-MMP knockout effects on the most abundant neural cell type in the human brain, namely, astrocytes. To evaluate astrocyte features independently of neurons, hiPSCs were differentiated using a previously published protocol [32], with minor modifications. For both genotypes, we observed heterogeneity in cell morphology of the astrocyte-like cultures. Using the previously published method [33], we determined and binned all cells into three morphological categories for further analysis: fibroblast-like, arborized, and arrowhead (Figure 5A). Detailed analysis showed differences in the proportion of astrocytes falling into each category between genotypes, suggesting that MT5-MMP deficiency could have an impact on astrocyte morphology (two-way repeated ANOVA with Sidak’s post hoc test, *N* = 3, *n* = 245, *F* (2, 8) = 38.7, *p* < 0.001). The comparison of morphotypes between genotypes showed that the proportion of “arborized” astrocytes was higher in wild-type astrocyte-like cultures (WT 53/131 vs. KO 14/114, *p* < 0.01), meanwhile, the number of fibroblast-like cells was higher in MT5-MMP-deficient cultures (WT 38/131 vs. KO 76/114, *p* < 0.05) (Figure 5B). A 3D IsoSurface reconstruction from serial confocal z-stacks of GFAP cells, as well as the quantification of cell surface area (SA), cell volume (Vol) and the SA:Vol ratio, revealed significant differences in cellular morphology (Figure 5C–E). Analysis of the genotype effects for each of the morphotypes (nested *t*-test, *N* = 3, *n* = 73; arborized, *n* = 33, *F* = 0.013, *p* = 0.92; arrowhead, *n* = 24, *F* = 12.5, ** *p* = 0.002; fibroblast-like, *n* = 16, *F* = 13.8, * *p* = 0.02) revealed that arrowhead and fibroblast-like MT5-MMP KO astrocytes were characterized by a bigger surface area in comparison to WT astrocytes of the same morphotype (Figure 5C). Quantification of cell volume revealed significant differences between genotypes (nested *t*-test, *N* = 3, *n* = 73; arborized: *n* = 33, *F* = 3.55, *p* = 0.13; arrowhead: *n* = 24, *F* = 13.8, *** *p* < 0.001; fibroblast-like: *n* = 16, *F* = 8.98, *p* = 0.04). MT5-MMP KO astrocytes were characterized by increased volume for arrowhead (*** *p* < 0.001) and fibroblast-like (* *p* < 0.05) astrocytes, in comparison to WT ones (Figure 4D). However, the SA:Vol ratio was higher for arborized WT compared to MT5-MMP KO astrocytes (nested *t*-test, *N* = 3, *n* = 73; arborized: *n* = 33, *F* = 9.60, ** *p* = 0.004; arrowhead: *n* = 24, *F* = 4.75, *p* = 0.09; fibroblast-like: *n* = 16, *F* = 4.08, *p* = 0.06, Figure 5E).

Astrocytes at the final stage of differentiation (45 DIV) exhibited the expression of astroglia-specific markers—GFAP, GLAST, ALDH1L1, and S100B (Figure 6A). Quantification of the fluorescence intensity of GFAP+ (nested *t*-test, *N* = 3, *n* = 245; arborized: *n* = 67, *F* = 3.92, *p* = 0.12; arrowhead: *n* = 64, *F* = 0.381, *p* = 0.57; fibroblast-like: *n* = 114, *F* = 0.433, *p* =0.55) and ALDH1L1+ cells (nested *t*-test, *N* = 3, *n* = 121; arborized: *n* = 36, *F* = 0.04, *p* = 0.85; arrowhead: *n* = 30, *F* = 1.86, *p* = 0.24; fibroblast-like: *n* = 55, *F* = 0.746, *p* = 0.44) revealed no difference in intensity between genotypes (Figure 6B,C). However, analysis of GLAST showed significantly decreased intensities for all three morphotypes of MT5-MMP KO astrocytes, compared to WT cells (nested *t*-test, *N* = 3, *n* = 181; arborized: *n* = 57, *F* = 31.2, ** *p* = 0.005; arrowhead: *n* = 32, *F* = 5.80, * *p* = 0.02; fibroblast-like: *n* = 92, *F* = 14.0, * *p* = 0.019, Figure 6D). Finally, analysis of S100β intensity showed significant difference between arborized MT5-MMP-deficient and wild-type astrocytes of the same morphotype (nested *t*-test, *N* = 3, *n* = 139; arborized: *n* = 49, *F* = 9.67, *p* = 0.04; arrowhead: *n* = 42, *F* =1.48, *p* = 0.07; fibroblast-like: *n* = 48, *F* = 0.05, *p* = 0.83, Figure 3E).

## 4. Discussion

In the present study, we demonstrated for the first time the importance of human MT5-MMP for the shape and physiology of neurons and astrocytes derived from hiPSCs. Innovative approaches based on hiPSC technologies help to bridge the genetic and phenotypic gap between animal models and humans for a better understanding of universal biological pathways, therapeutic target discovery and disease modeling [34,35,36].

We found that MT5-MMP-deficient neurons exhibited more branched morphology compared to wild-type neurons. In line with our data, Folgueras et al. showed that DRG-neurons from MT5-MMP knockout neonatal mice exhibited an increase in neurite ramification and varicosities compared to wild-type neurons, as well as in the number of branching points and in the total number of secondary neurites per neuron [24]. These data support the idea that MT5-MMP is pivotal in the developmental process of both CNS and PNS neurons. Neuritogenesis is a highly dynamic process in which N-cadherin, a substrate of MT5-MMP, participates in the stabilization of neurite branches [37]. Therefore, changes in N-cadherin processing and distribution might modulate the branched morphology of neurons. However, despite the fact that N-cadherin is an integral part of neural synapses [38,39], we did not observe any gross abnormalities in synaptic development between genotypes in a context where the electrophysiological analysis revealed the immature state of neurons. Moreover, both wild-type and MT5-MMP-deficient neurons displayed a low density of double-positive synaptic puncta even after 8 weeks *in vitro*, which further emphasizes the relative immaturity of cells. Perhaps, for this reason, the full phenotype of MT5-MMP-deficient cells in physiological conditions is not obvious at the studied developmental stage. Trial analysis (unpublished data) of vGluT1 and PSD95 showed that the number of vGluT1 positive puncta was low, even at day 60. PSD95 was detectable at early stages, i.e., at day 25–30 DIV, the expression peaked on DIV40 and remained on the same level up to DIV60, when the cultures were used for morphological analysis. In line with that, we speculate that MT5-MMP deficiency might affect axonal differentiation, as was previously demonstrated in primary neuronal cultures from E18 mouse embryos [22]. This hypothesis should be confirmed in future works with more mature human neurons derived from hiPSCs.

Astrocytes are fundamental components of the CNS. They constitute the most abundant cell type in the human brain and exhibit various morphologies, whose correlation with functional phenotypes is not yet well understood [33,40,41]. Neuron–astrocyte interactions are critical for normal neuronal functioning since astrocytes are involved in the guidance of neurons, and dynamically modulate synaptic transmission as part of the tripartite synapse [42,43,44]. Hence, we aimed to produce and analyze astrocytes derived from MT5-MMP knockout human iPSCs. Analysis of the fluorescence intensities of astrocytic markers, as well as of morphological features, revealed differences between wild-type and MT5-MMP KO astrocytes. High levels of GFAP mRNA and protein are recognized as a prominent feature of reactive astrocytes, although they may also reflect adaptive responses to physiological stimuli [45,46]. However, in our experiments, we did not observe a significant difference in GFAP fluorescence intensity for all morphotypes, possibly indicating a basal activation level of these cells, which is indistinguishable from the wild-type one. However, GFAP is not an absolute marker of astrocyte phenotypic and functional diversity [46]. That is why we conducted the analysis for other astrocytic markers, among which S100β is also a marker of reactivity [45,47] that turned out to be downregulated in MT5-MMP KO astrocytes. The downregulation of S100β in human astrocytes might explain an anti-inflammatory effect of MT5-MMP deficiency, which was previously shown for the 5xFAD mouse model of AD lacking MT5-MMP [25,26]. Particularly, 5xFAD/MT5-MMP KO animals showed significant reductions in key inflammatory mediators, such as interleukin-1 beta (IL-1β) and tumor necrosis factor alpha (TNFα), as well as reduced astrocyte and microglia reactivity. Since reduced neuroinflammation was concomitant with strong decreases in Aβ levels, we inferred a sequence of events where MT5-MMP deficiency caused an Aβ reduction and subsequently less inflammation. Our current data suggest that MT5-MMP can also control inflammation per se in a non-pathological state in a human setting, but it remains an open question whether basal downregulation of inflammatory markers predisposes astrocytes to decreased reactivity. In any case, our results reinforce the idea that MT5-MMP is not only an APP-cleaving enzyme but also that its scope of activity goes far beyond that [12,13,14].

We found the expression of glutamate transporter GLAST is downregulated in all three morphotypes. Previously, it was shown that reductions in GLAST expression have been associated with a limited ability of astrocyte glutamate uptake from the extracellular space [48,49]. Trial co-culturing experiments showed that MT5-MMP knock-out astrocytes could not support neuronal viability, resulting in massive cell death (unpublished data). We might infer that MT5-MMP deficiency impairs the ability of astrocytes to maintain extracellular physiological levels of glutamate and, therefore, their ability to protect neurons from excitotoxicity. The verification of this hypothesis should warrant further study in neuron-astrocyte co-cultures. On the other hand, the stability of the metabolic enzyme ALDH1L1 between astrocytes of both genotypes possibly indicates the lack of MT5-MMP effects on metabolic functions in these cells.

Analysis of astrocyte morphology also revealed differences between genotypes. The round shape fibroblast-like morphotype prevailed among MT5-MMP knockout astrocytes. In addition, MT5-MMP KO astrocytes were characterized by increased surface area and volume, indicating a more “swallow” morphology. Meanwhile, the surface area/volume ratio was higher for arborized wild-type cells that again indicates a more rounded morphology of arborized MT5-MMP KO cells. We consider two possible hypotheses to explain this phenomenon. The first concerns the ability of MT5-MMP to convert by proteolysis the inactive pro-MMP-2 into the MMP-2 active form [50,51], and the fact that MMP-2 is a key enzyme for astrocyte migration and morphology [52]. Accordingly, Ogier and colleagues demonstrated, in an *in vitro* model of 2D astrocyte migration, that a selective MMP-2 inhibitor induced a round-like cell shape, coincident with impaired cell motility. In contrast, untreated cells exhibited a more elongated shape with stress fibers, filopodia and lamellipodia at the leading edge of migration [52]. Additionally, MMP-2 is considered as an enhancer of the neuroinflammatory response through the regulation of cytokine activity and release, notably in astrocytes [53]. Thus, taking into account these data, we suggest that MT5-MMP knockout astrocyte morphology may be, at least in part, the result of MMP-2 processing abnormalities that eventually downregulate inflammation.

The second hypothesis concerns the possibility that MT5-MMP is involved in astrocyte maturation, if we take into account the fact that MT5-MMP KO astrocytes are characterized by the downregulation of S100β. In particular, it was previously shown that rat primary astrocytes transiently downregulated S100β expression when exposed to the differentiating agent, db-cAMP, and re-expressed S100β at later stages of db-cAMP-induced differentiation [54,55]. Therefore, simpler morphology and a decreased expression of S100b could be the result of a developmental lag.

Overall, our study provides the first experimental evidence that MT5-MMP may affect some of the phenotypical features of cultured human neurons and astrocytes, which are reminiscent of cells at early developmental stages. This places MT5-MMP as an important enzyme in the control of neural cell development and warrants future work to identify more precisely the molecular mechanisms at stake in physiological and pathological settings.

## Figures and Tables

**Figure 1 cells-10-01705-f001:**
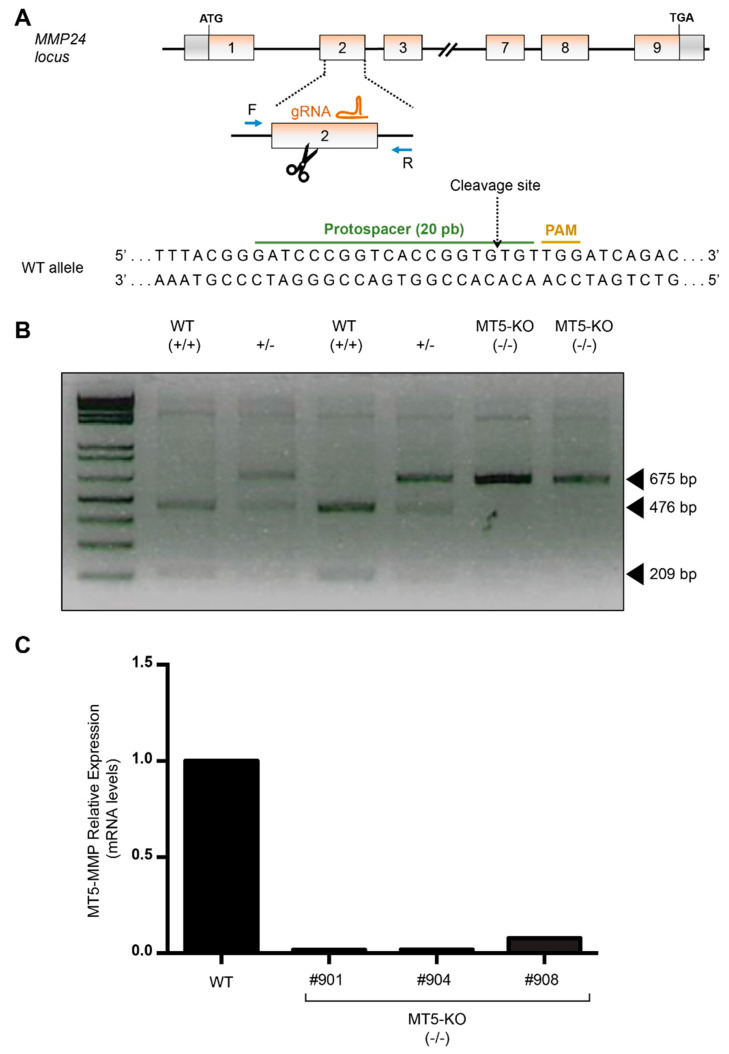
MT5-MMP (*MMP24*)-KO hiPSC generation by CRISPR/Cas9 genome editing. (**A**) Schematic representation of the wild-type (WT) locus and the genome editing strategy used to generate the knockout (KO) isogenic clones from a non-demented donor hiPSC line. (**B**) Representative PCR-based screening for the editing of *MMP24*: WT (+/+), heterozygous (+/−) and homozygous MT5-KO (−/−) clones. The F and R primers for *MMP24* were designed flanking the Cas9-cutting site. The PCR product was digested and run on an electrophoresis gel, to analyze the size of the amplicon for each individual clone (expected profiles: two bands at 476 bp and 209 bp (WT +/+); one band at 675 bp (restriction enzyme site obliterated, MT5-KO -/-) and three bands at 675bp, 476bp and 209bp (heterozygous clones, +/−). (**C**) After Sanger sequencing validation of MT5-KO clones (not shown), qPCR analysis revealed that *MMP24* mRNA was barely detectable in the newly generated MT5-KO clones.

**Figure 2 cells-10-01705-f002:**
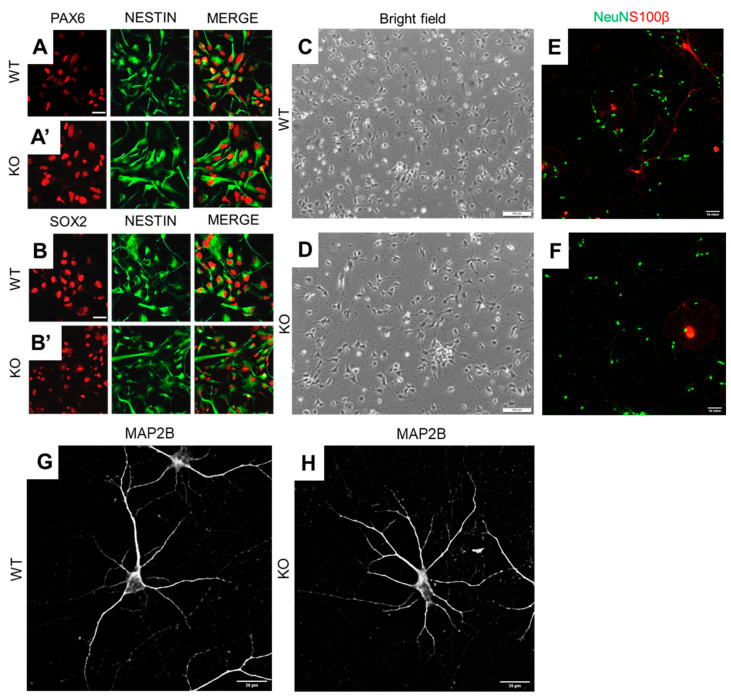
Efficient derivation of NPCs and neurons from wild-type and MT5-MMP-deficient hiPSCs. (**A**,**B**) Wild-type and MT5-MMP-deficient (KO) hiPSCs were efficiently differentiated into NPCs co-expressing the paired box 6 (PAX6, red) and nestin (green) (**A**,**A’**), as well as the SRY (sex-determining region Y)-box 2 (SOX2, red) and Nestin (green) markers (**B**,**B’**). Scale bar 20 µm. (**C**,**D**) Representative bright-field images of wild-type (**C**) and MT5-MMP-deficient NPCs (**D**) undergoing neuronal differentiation at DIV3, scale bar 50 µm. Note that multiple neurite-like protrusions were already observed at this early stage of differentiation, with no clearly distinct morphology between neurons of both genotypes at this stage. (**E–H**) Representative images of immunolabeled hiPSC-derived neuronal cultures immunolabelled with antibodies against the neuron-specific marker NeuN (green) and the astrocyte-specific marker S100β (red) (**E**,**F**, scale bar 50 µm) after 60 days of differentiation (DIV60). In addition, cells expressed neuron-specific cytoskeletal protein MAP2B (**G**,**H**, scale bar 20 µm).

**Figure 3 cells-10-01705-f003:**
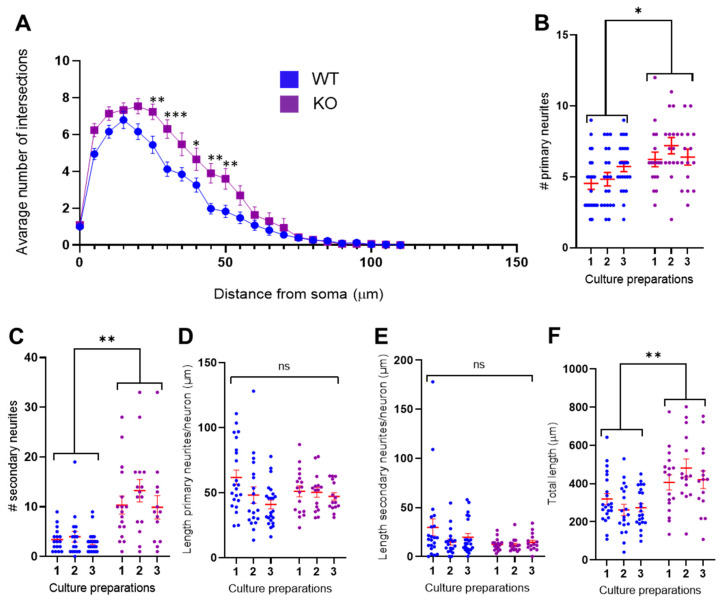
MT5-MMP deficiency alters neurite outgrowth in hiPSC-derived neurons. NPCs derived from wild-type and MT5-MMP KO hiPSCs were differentiated into neurons for 60 days before being subjected to the Sholl analysis. (**A**) Sholl analysis of dendrites from neurons of both genotypes (as indicated) showing the total number of intersections of MAP2B positive neurites with concentric rings, from the cell body to distal ends of dendrites (maximal distance of 110 μm, steps of 5 μm). MT5-MMP deficient neurons showed higher dendritic intersections in the 30 to 50 µm distance from soma (two-way repeated-measures ANOVA with Bonferroni’s post hoc test, *N* = 3, *n* = 82, *F* (1, 80) = 10.7, *** *p* < 0.001; ** *p* < 0.01; * *p* < 0.05). (**B**,**C**) Bar charts depicting the counting of the number of primary (**B**) and secondary (**C**) neurites using the NeuronJ plugin. Noticeably, analyses showed that MT5-MMP deficiency significantly increases the number of primary (nested *t*-test, *N* = 3, *n* = 111, *F* = 10.0, * *p* = 0.03) and secondary neurites (nested *t*-test, *N* = 3, *n* = 111, *F* = 42.8, ** *p* = 0.003). (**D**,**E**) Bar charts depicting the average length of primary (**D**) and secondary (**E**) neurites. (**F**) Bar chart showing a significant increase of total length for MT5-MMP knockout neurons (nested *t*-test, *N* = 3, *n* = 111, *F* = 28.0, ** *p* = 0.006). Data are presented as mean ± SEM.

**Figure 4 cells-10-01705-f004:**
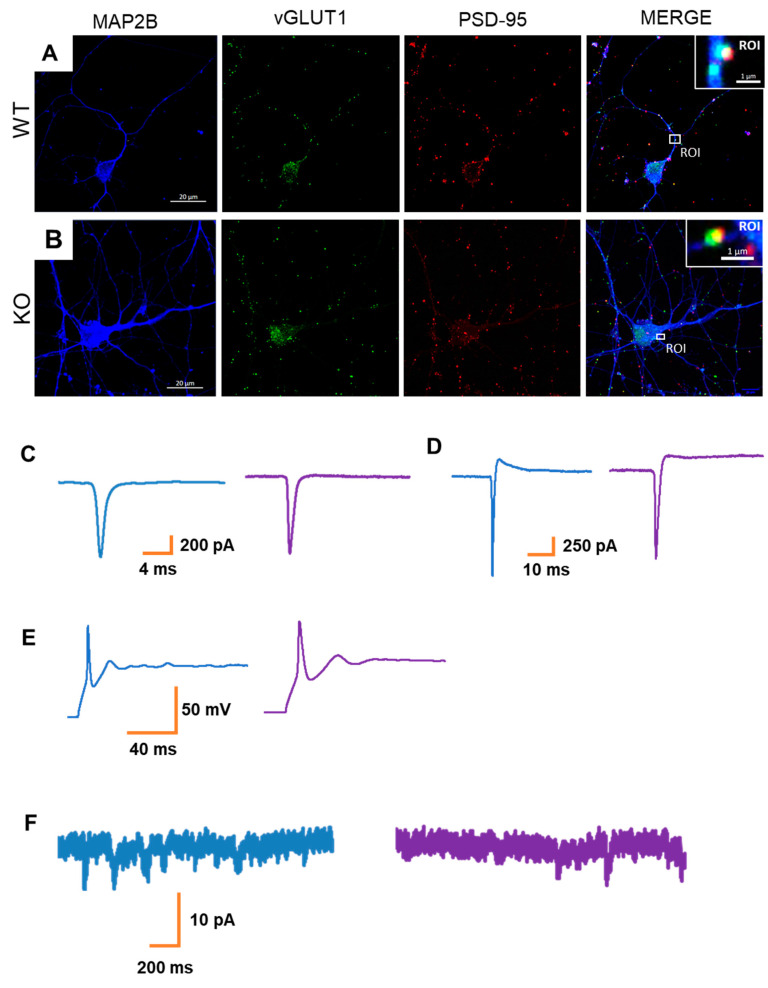
Synaptic connectivity and functionality of hiPSC-derived NPCs. (**A**,**B**) Representative images of wild-type (**A**) and MT5-MMP-deficient (**B**) neuronal cultures at 8 weeks of differentiation. The cells were immunostained by antibodies against MAP2B (blue), vGluT1 (green), and PSD-95 (red) as dendritic, pre- and postsynaptic markers, respectively. Scale bar 20 µm. The white rectangle depicts the ROI. Note that the juxtaposed position of pre- and postsynaptic proteins, vGluT1 and PSD95, characterized by a small area of overlapping colors, was observed for both genotypes. Scale bar 1 µm. (**C**,**D**) Representative traces from whole-cell patch-clamp recording showing that neurons were characterized by small Na^+^ and K^+^ currents at 60DIV. (**E**) Representative traces showing that neurons at 60DIV exhibited single action potentials in response to a depolarization step. (**F**) Representative traces from recorded neurons at 60DIV showed small inward spontaneous currents.

**Figure 5 cells-10-01705-f005:**
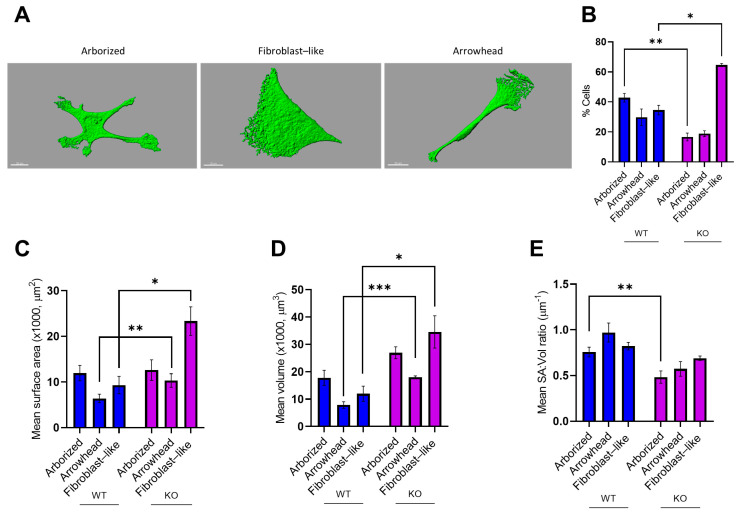
MT5-MMP-deficient astrocyte-like cells display morphological changes. (**A**) Representative images obtained with 3D IsoSurface reconstruction from serial confocal z-stacks of GFAP-stained astrocytes, showing clear differences in morphology, were binned into three categories: arborized, fibroblast-like, and arrowhead. (**B**) Bar chart showing the percentage of cells belonging to each of the three morphotypes from wild-type and MT5-MMP knockout (KO) astrocyte cultures. Note that MT5-MMP KO cultures showed a greater proportion of fibroblast-like astrocytes at the expense of arborized astrocytes (two-way ANOVA with Sidak´s post hoc test, *N* = 3 culture preparations, *n* = 245 GFAP+ astrocytes, *F* (2, 239) = 7.72, ** *p* < 0.01; * *p* < 0.05). (**C**–**E**) Bar charts depicting the comparative quantification of astrocyte-like cell surface area (nested *t*-test, *N* = 3, *n* = 73; arborized, *n* = 33, *F* = 0.013, *p* = 0.92; arrowhead, *n* = 24, *F* = 12.5, ** *p* = 0.002; fibroblast-like, *n* = 16, *F* = 13.8, * *p* = 0.02, in (C), volume (nested *t*-test, *N* = 3, *n* = 73; arborized: *n* = 33, *F* = 3.55, *p* = 0.13; arrowhead: *n* = 24, *F* = 13.8, *** *p* < 0.001; fibroblast-like: *n* = 16, *F* = 8.98, *p* = 0.04, in (D) and SA:Vol ratio (nested *t*-test, *N* = 3, *n* = 73; arborized: *n* = 33, *F* = 9.60, ** *p* = 0.004; arrowhead: *n* = 24, *F* = 4.75, *p* = 0.09; fibroblast-like: *n* = 16, *F* = 4.08, *p* = 0.06, in (E) between the different genotypes. Data are presented as mean ± SEM.

**Figure 6 cells-10-01705-f006:**
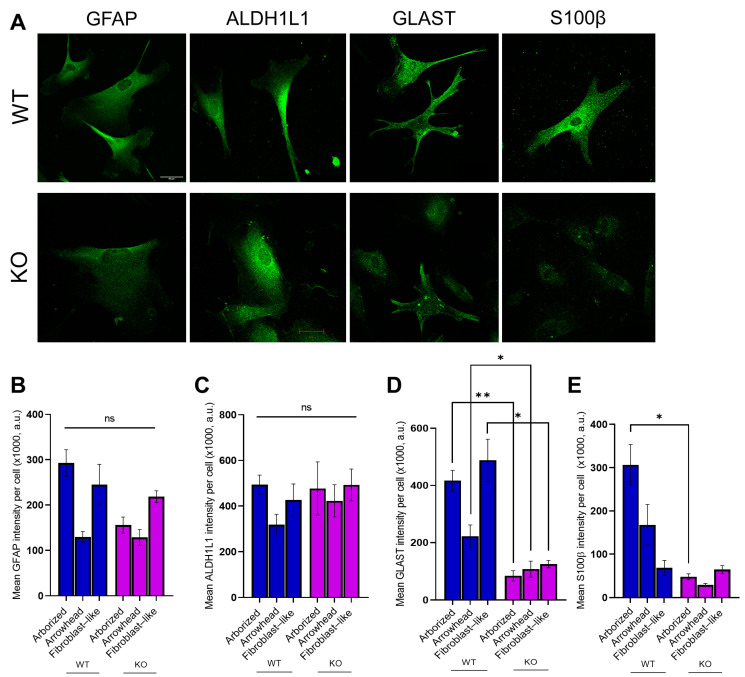
MT5-MMP-deficient astrocyte-like cells display altered expression of several astrocytic markers. (**A**) Representative images of hiPSC-derived astrocyte-like cells from wild-type and MT5-MMP KO hiPSCs, immunostained at DIV 45 for different astrocytic markers as indicated (scale bar: 20 µm). (**B**–**E**) Bar charts showing the quantification of the mean fluorescence intensity for different astrocytic markers within each morphological category. MT5-MMP-deficient astrocytes were characterized by no difference for GFAP (nested *t*-test, *N* = 3, *n* = 245; arborized: *n* = 67, *F* = 3.92, *p* = 0.12; arrowhead: *n* = 64, *F* = 0.381, *p* = 0.57; fibroblast-like: *n* = 114, *F* = 0.433, *p* = 0.55, in (**B**), ALDH1L1 (nested *t*-test, *N* = 3, *n* = 121; arborized: *n* = 36, *F* = 0.04, *p* = 0.85; arrowhead: *n* = 30, *F* = 1.86, *p* = 0.24; fibroblast-like: *n* = 55, *F* = 0.746, *p* = 0.44, in (**C**). Meanwhile, the significant difference was observed for GLAST (nested *t*-test, *N* = 3, *n* = 181; arborized: *n* = 57, *F* = 31.2, ** *p* = 0.005; arrowhead: *n* = 32, *F* = 5.80, * *p* = 0.02; fibroblast-like: *n* = 92, *F* = 14.0, * *p* = 0.019, in (**D**). Analysis of S100β fluorescence intensity (nested *t*-test, *N* = 3, *n* = 139; arborized: *n* = 49, *F* = 9.67, *p* = 0.04; arrowhead: *n* = 42, *F* = 1.48, *p* = 0.07; fibroblast-like: *n* = 48, *F* = 0.05, *p* = 0.83, in (**E**) reveals a significant difference between the arborized morphotype of WT and MT5-MMP KO astrocytes. Data are presented as mean ± SEM.

## Data Availability

The data presented in this study are openly available in Mendeley Data at DOI:10.17632/bfj8pnxfh8.1.

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
