# Peer review of "Deficiency in MT5-MMP Supports Branching of Human iPSCs-Derived Neurons and Reduces Expression of GLAST/S100 in iPSCs-Derived Astrocytes"

_cells, 2021, doi:10.3390/cells10071705_

Round 1

Reviewer 1 Report

Review of a manuscript “Deficiency in MT5-MMP supports branching of human iPSCs-2derived neurons and reduces expression of GLAST/S100 in iP-3SCs-derived astrocytes” by Nikita Arnst and coauthors

Alzheimer’s disease is the most prevalent neurodegenerative disease for which no disease modifying therapy is developed. Abnormal accumulation of amyloid β peptide is a hallmark of this disorder. The course of the disease is controlled by several key enzymes, involved in the processing of amyloids, i.e. β-site APP cleaving enzyme 1 (BACE1) and γ-secretase.  More recent findings revealed an important role of the Membrane-Type 5 Matrix Metalloproteinase (MT5-MMP/MMP-24) in APP-processing. The authors studies a role of MT5-MMP in human neurons and astrocytes and their potential functional crosstalk in a normal and pathological states. They made an important conclusion about the importance of human MT5-MMP for the shape and physiology of neurons and astrocytes derived from hiPSCs. The topic of this study is important and the results will be interesting for the readers of “Cells”.

The following corrections should be made.

Abstract

Line 19: “However, a more comprehensive analysis on the physiological role of MT5-MMP is necessary to evaluate how its targeting affect neural cells in non-pathological situations.” It is not clear why the authors narrow here the aims to “neural cells in non-pathological situations”. Their results are also related to pathological conditions and to astrocytes.  This sentence should be modified to become more relevant to the overall content of the manuscript.

Introduction

Lines 42-43: After the sentence “However, the high failure rate of clinic-step BACE1 inhibitor candidates and its interference with the physiological processing of other substrates stimulates searching for novel therapeutic targets in AD [8, 9, 10].” The authors should add the following introductory sentence:” Matrix metalloproteinases (MMPs) - enzymes controlling intracellular and extracellular amyloidogenic proteins turnover matrix are associated with neurodegenerative diseases [ref. Surguchev et al. , Cell Responses to Extracellular α-Synuclein. Molecules. 2019 Jan 15; 24(2):305.]

Lines 73-74: “Besides, confocal imaging and qRT-PCR analyses of samples after traumatic brain injury (TBI) and bilateral entorhinal cortical lesion reveled that MT5-MMP inhibition attenuated ADAM-10 and increased N-cadherin [18].” The authors of ref. 18 observed time dependent changes of N-cadherin, and at 2 and 7 days it was reduced. So it would be more correct to write “… attenuated ADAM-10 and N-cadherin [18].”

Lines 81-83: ”MT5-MMP deficiency led to changes in neuron and astrocyte morphology, characterized by the formation of more branched neurons and round-shaped astrocytes with decreased expression of S100b and GLAST. These data corroborate the multifaceted spectrum of MT5-MMP actions in neural cells.”

There is contradiction between the first and the second sentences. In the first the authors state changes in both neurons and astrocytes, in the second they conclude that the changes occur in neural sentence. Should be clarified.

Materials and Methods

Line 118: “2.4. CRISPR-Cas9-mediated edition for the generation of MMP24 knockout hiPSCs

This chapter is overloaded with details. It may be made more concise with references to previously published results.

Line 186: For neuronal differentiation, hiPSC-derived NPCs were differentiated” the sentence should be corrected as follows: ”hiPSC-derived NPCs were differentiated…”

Results

Figure 1B

An arrow showing expected band with 209 bp is located at a higher position than real bands in WT/WT (+/+)

Author Response

Reviewer 1

Alzheimer’s disease is the most prevalent neurodegenerative disease for which no disease modifying therapy is developed. Abnormal accumulation of amyloid β peptide is a hallmark of this disorder. The course of the disease is controlled by several key enzymes, involved in the processing of amyloids, i.e. β-site APP cleaving enzyme 1 (BACE1) and γ-secretase.  More recent findings revealed an important role of the Membrane-Type 5 Matrix Metalloproteinase (MT5-MMP/MMP-24) in APP-processing. The authors studies a role of MT5-MMP in human neurons and astrocytes and their potential functional crosstalk in a normal and pathological states. They made an important conclusion about the importance of human MT5-MMP for the shape and physiology of neurons and astrocytes derived from hiPSCs. The topic of this study is important and the results will be interesting for the readers of “Cells”.

The following corrections should be made.

Abstract

Line 19: “However, a more comprehensive analysis on the physiological role of MT5-MMP is necessary to evaluate how its targeting affects neural cells in non-pathological situations.” It is not clear why the authors narrow here the aims to “neural cells in non-pathological situations”. Their results are also related to pathological conditions and to astrocytes.  This sentence should be modified to become more relevant to the overall content of the manuscript.

Reply: We appreciate the comments of reviewer. We changed the sentence to: “However, a more comprehensive analysis on the role of MT5-MMP is necessary to evaluate how its targeting affects neurons and glia in pathological and physiological situations.”  Formally, the term “neural”, in contrast to “neuronal” is very general and includes all cells of the nervous system, i.e. both neurons and astrocytes studied here. However, to increase clarity, we replaced it by “neurons and glia”. We provide a version of manuscript with all changes being underlined for convenience of reviewers.

Introduction

Lines 42-43: After the sentence “However, the high failure rate of clinic-step BACE1 inhibitor candidates and its interference with the physiological processing of other substrates stimulates searching for novel therapeutic targets in AD [8, 9, 10].” The authors should add the following introductory sentence:” Matrix metalloproteinases (MMPs) - enzymes controlling intracellular and extracellular amyloidogenic proteins turnover matrix are associated with neurodegenerative diseases [ref. Surguchev et al. , Cell Responses to Extracellular α-Synuclein. Molecules. 2019 Jan 15; 24(2):305.]

Reply: We added the requested introductory sentence and the reference.

Lines 73-74: “Besides, confocal imaging and qRT-PCR analyses of samples after traumatic brain injury (TBI) and bilateral entorhinal cortical lesion reveled that MT5-MMP inhibition attenuated ADAM-10 and increased N-cadherin [18].” The authors of ref. 18 observed time dependent changes of N-cadherin, and at 2 and 7 days it was reduced. So it would be more correct to write “… attenuated ADAM-10 and N-cadherin [18].”

Reply: We introduced the requested change of the text.

Lines 81-83: ”MT5-MMP deficiency led to changes in neuron and astrocyte morphology, characterized by the formation of more branched neurons and round-shaped astrocytes with decreased expression of S100b and GLAST. These data corroborate the multifaceted spectrum of MT5-MMP actions in neural cells.”

There is contradiction between the first and the second sentences. In the first the authors state changes in both neurons and astrocytes, in the second they conclude that the changes occur in neural sentence. Should be clarified.

Reply: We have changed the concluding sentence to: These data corroborate the multifaceted spectrum of MT5-MMP actions in the nervous system.

Materials and Methods

Line 118: “2.4. CRISPR-Cas9-mediated edition for the generation of MMP24 knockout hiPSCs

This chapter is overloaded with details. It may be made more concise with references to previously published results.

Reply: Following the advice of reviewer, we added a reference and shorten this chapter.

Line 186: For neuronal differentiation, hiPSC-derived NPCs were differentiated” the sentence should be corrected as follows: ”hiPSC-derived NPCs were differentiated…”

Reply: The requested change was introduced.

Results

Figure 1B

An arrow showing expected band with 209 bp is located at a higher position than real bands in WT/WT (+/+)

Reply: We modified the figure to be more precise.

Reviewer 2 Report

The manuscript is clearly written and presented. Furthermore, It provides relevant information that deserves to be published. I do not have critics and I would accept it for publication in the present form.

Author Response

Reviewer 2

The manuscript is clearly written and presented. Furthermore, It provides relevant information that deserves to be published. I do not have critics and I would accept it for publication in the present form.

Reply: We appreciate the time and this evaluation of the reviewer.